# Improving Sexual Function by Using Focal Vibrations in Men with Spinal Cord Injury: Encouraging Findings from a Feasibility Study

**DOI:** 10.3390/jcm8050658

**Published:** 2019-05-11

**Authors:** Rocco Salvatore Calabrò, Antonino Naro, Massimo Pullia, Bruno Porcari, Michele Torrisi, Gianluca La Rosa, Alfredo Manuli, Luana Billeri, Placido Bramanti, Fabrizio Quattrini

**Affiliations:** 1Department of Robotic Neurorehabilitation, IRCCS Centro Neurolesi Bonino Pulejo, 98124 Messina, Italy; g.naro11@alice.it (A.N.); massimo.pullia@irccsme.it (M.P.); bruno.porcari@irccsme.it (B.P.); michele.torrisi@irccsme.it (M.T.); gianluca.larosa@irccsme.it (G.L.R.); alfredo.manuli@irccsme.it (A.M.); luana.billeri@irccsme.it (L.B.); dino.brama@gmail.com (P.B.); 2Department of Applied Clinical Sciences and Biotechnology, University of L’Aquila, 67100 L’Aquila, Italy; fquattrini@gmail.com

**Keywords:** sexual dysfunction, spinal cord injury, focal muscles vibrations, international index of erectile function

## Abstract

Erectile dysfunction (ED) is a frequent and disabling condition in patients with spinal cord injury (SCI). Spasticity can negatively affect sexual intercourse, as it may interfere with positioning, mobility, and muscle activation and strength, leading to ED. The aim of our study was to evaluate the feasibility and efficacy of muscle vibration (MV) applied to the pelvic muscles in improving ED in men with SCI. Ten adult men with traumatic SCI were submitted to 15 sessions of MV, applied on the perineum and the suprapubic and sacrococcygeal areas, using a pneumatic vibrator. MV was performed three times a week for five consecutive weeks, each session lasting 30 min. Muscle tone and sexual function were assessed before and after MV using the Modified Ashworth Scale (MAS) and International Index of Erectile Function (IIEF). We assessed the cremasteric and bulbocavernosus reflexes, as well as the electrophysiological bulbocavernosus reflex (eBCR) and pudendal nerve somatosensory-evoked potential (PSEP). MV was safe and well tolerated. All the patients reported an improvement in MAS and IIEF, with better reflexive responses, and a significant increase in eBCR and PSEP amplitude. In conclusion, MV of the pelvic floor is a promising method to reduce segmental spasticity and improve ED in men with incomplete SCI. However, our findings require confirmation through a randomized clinical trial with a larger sample size and longer trial period to examine long-term after effects.

## 1. Introduction

Spinal cord injury (SCI) can severely impair erection, ejaculation, and perception of orgasm, with a consequent change in patients’ sexual behavior [1,2]. About 20% of patients with complete SCI complain of severe erectile dysfunction (ED), i.e., the inability to achieve or maintain an erection sufficient for satisfactory sexual performance [1,2,3]. ED severity varies in men with incomplete SCI, depending on the location and degree of damage of the neural pathways subtending erection [1,2,3].

Erection is a neurovascular event characterized by the tumescence of the cavernous bodies that rely upon integration of neural and humoral mechanisms at various levels of the nervous system. It requires the participation of autonomic and somatic (i.e., the dorsal nerve of the penis) nerves and the integration of numerous spinal and supraspinal sites, including the medial preoptic area of the hypothalamus, and septal region (Figure 1). The penis receives innervation from sacral parasympathetic (pelvic), thoracolumbar sympathetic (hypogastric and lumbar chain), and somatic (pudendal) nerves. The major excitatory input to the penis is provided by the parasympathetic nervous system (S2–S4), responsible for vasodilatation of the penile vasculature and erection. The sympathetic nervous system (T10–L2) seems to play a pivotal role in the detumescence of the penis [4].

SCI may cause structural changes within the pelvic floor, neuropathic pain, spasticity, and problems with bladder and bowel continence with the consequent shame of unwanted smells. Altogether, these concerns can worsen ED significantly. Changes in body image representation, depression, anxiety, and fatigue also contribute to worsen ED [5,6].

Whether achieved, reflexogenic, psychogenic, or mixed, erection is often short lasting and non-sufficient to maintain satisfactory sexual relations [5,6]. Frequency and quality of sexual activity inevitably decline after SCI. This obviously represents a serious concern in people with SCI, as sexual activity is an important factor in quality of life and self-esteem [5,6]. Diminished or absent sexual activity does not mean absent sexuality, as the need for sexual expression and intimacy remains [7].

Most of the patients with SCI require specific, multidisciplinary management for ED, including pharmacological and surgical approaches, counseling, and rehabilitation with physical therapy treatment [4,8,9]. For instance, the oral administration of PDE-5 inhibitors represents one of the best choices in men with SCI, if the basic mechanisms responsible for erection, including both normal vasculature and S2–S4 reflex arc, are preserved [10]. Injectable and intra-urethral agents are relegated to second line therapy after oral PDE-5 inhibitors are ineffective, although the local delivery of medications (i.e., PGE1 and papaverine) remains useful in about 25%–30% of ED patients. Penile prosthesis offers a valid therapeutic alternative for patients who fail vasoactive drugs and vacuum-constrictive devices and who are not candidates for vascular reconstruction procedures [4]. However, such approaches may be unacceptable to the patient, unavailable, or inapplicable [4,8,9].

Growing research is focusing on instrumental approaches to manage ED. It is worth noting that among the problems facing patients with SCI, spasticity significantly contributes to ED [11,12,13,14]. Spasticity develops gradually over several months after SCI, is strictly associated with hyperreflexia and excessive muscle spasms, and involves also pelvic floor muscles [11,12,13,14]. Thus, spasticity may affect: (1) positioning and mobility that are necessary for sexual intercourse, and (2) activation, strength, time of force development, and control of the pelvic floor muscles that are involved in erectile and ejaculation functions, with spasms during penetration [11,12,13,14]. Therefore, the reduction of spasticity may be a therapeutic option to improve ED.

A useful tool to reduce segmental spasticity is represented by muscle vibration (MV), as suggested by reports on patients with stroke, multiple sclerosis, and cerebral palsy [15,16,17,18]. Vibrations have been already used to manage different pelvic floor dysfunctions due to diverse pathologies, but only employing whole body vibration (not addressing ED) or penile vibration used for ejaculation [19,20,21,22,23,24,25,26,27]. There is converging evidence that MV provides the central nervous system with strong proprioceptive inputs (preferentially through Ia afferents) that reach the somatomotor cortices. This may help to modify the corticospinal excitability, to favor intracortical inhibitory systems, and to induce better muscle synergy pattern by acting on the excitability of spinal motoneurons and interneurons. MV may directly act at the spinal level, reducing abnormalities of the spinal excitability (as the activation of primary Ia spindle afferents result in inhibition of the monosynaptic reflex) and restoring abnormal reciprocal and presynaptic inhibition mechanisms [15,16,17,18]. Altogether, these effects lead to a reduction in spasticity [15,16,17,18].

Hypothesizing that the MV-induced reduction of pelvic floor spasticity may improve ED, we evaluated the feasibility and efficacy of low-amplitude/high-frequency MV delivered to specific muscles of the pelvic floor and sacral area in improving ED in males with incomplete SCI.

## 2. Materials and Methods

### 2.1. Study Population

We screened SCI outpatients visiting the Behavioral and Robotic Neurorehabilitation Unit of our institute between January and June 2018 for patients complaining of ED. Ten patients with neurogenic ED were enrolled in this open label pilot study, according to the following inclusion criteria: (1) age 18–65 years; (2) diagnosis of incomplete SCI, with American Spinal Injury Association (ASIA) level C-D, no involvement of sacral spinal segment and SCI occurred at least six months before the study enrollment (i.e., chronic phase); (3) an International Index of Erectile Function (IIEF) score <18 (that is, severe, moderate, or mild-to-moderate ED) [28,29,30]; (4) to be sexually active and fully continent before SCI; and (5) withdrawal from previous treatment (for at least three months before the study inclusion), such as erectogenic aids, nutraceutics, antispastics, and psychoactive drugs.

### 2.2. Intervention

The patients were assessed at baseline using clinical and electrophysiological tests. MV was performed three times a week for five consecutive weeks, each session lasting 30 min. The stimulation was delivered to the perineum, suprapubic, and sacrococcygeal areas using a pneumatic vibrator powered by compressed air (Vibraplus, @-Circle; San Pietro in Casale, Bologna, Italy). The device is equipped with cup-like probes of 2 cm^2^ fixed to the suprapubic and sacrococcygeal areas by a Velcro strap, and a pen-like probe held by the therapist who carried out MV over the perineum. (Figure 2 and Figure 3).

Vibration stimuli were delivered at a frequency (f) of 150 Hz, with an amplitude (A) (i.e., of the peak-to-peak sinusoidal displacement of the underneath structures; on average 4 ± 0.5 mm) sufficient to evoke a progressive contraction of the perineal muscles in each of the participants, as assessed by the therapist who carried out the MV. The resulting intensity of stimulation was calculated by the formula (2πf) 2A.

### 2.3. Outcome Measures

Changes in IIEF (the primary end-point) were used to estimate the effectiveness of MV in improving ED. The IIEF [28,29,30] consists of 15 questions aimed at highlighting ED and its effects on the sex life in the 4 weeks preceding the test. The questionnaire evaluates 5 different fields of male sexuality within the different cultures. A: Erectile Function (Q1–5; 15, score 1–30), B: Orgasmic Function (Q9, 10; score 0–10), C: Sexual Desire (Q11, 12; score 2–10), D: Intercourse Satisfaction (Q6–8; score 0–15), and E: overall Satisfaction (Q13, 14, score 2–10). The total score ranges from 5–75, and is interpreted as follows: <10 severe, 11–16 moderate, 17–21 mild-to-moderate, 22–25 mild ED, and 26–30 no dysfunction.

The secondary end-points consisted in the improvement in pelvic floor muscle Modified Ashworth Scale (MAS), pelvic floor pain rated on a Visual Analogue Scale (VAS), groin and penile sensation, cremasteric and bulbocavernosus (BCR) reflexes, electrophysiological bulbocavernosus reflex (eBCR), and pudendal nerve somatosensory-evoked potential (PSEP).

MAS measures resistance during passive soft-tissue stretching and is used as a simple measure of spasticity (score 0: no tone–4: complete rigidity). VAS is a measurement instrument that tries to measure a characteristic (such as pain) or attitude that is believed to range across a continuum of values and cannot easily be directly measured (score 0: no pain–10: worst pain).

Before MAS and VAS administration, the patient underwent a complete physical examination, including general appearance, secondary sexual characteristics, cardiovascular system (blood pressure, peripheral pulses), genitourinary system, penis inspection (circumcision, deformity, plaques, phimosis, hypoesthesia), testes examination (size and consistency), rectal examination (sphincter tone and prostate examination), and groin and penile sensation (using Semmes–Weinstein monofilament sensory testing). To this end, the penis was sectored in six parts: the right and left halves of the shaft, the glans, the scrotum, and the left and right groin. The patient was blindfolded and each region was stimulated 3 times at random intervals. If all of the stimuli were experienced all 3 times correctly in all of the sectors, we concluded the patient had sensation.

The peripheral and central neural pathways underlying exteroceptive stimulation of the genital regions, the sacral spinal reflex mechanisms, the pudendal somatosensory and somatomotor pathways were studied using clinical (cremasteric reflex, BCR) and electrophysiological tests (eBCR, PSEP).

The cremasteric reflex is a superficial reflex found in human males that is related to emission and psychogenic erection (whereas BCR is more related to reflex erection). It is elicited when the inner part of the thigh is stroked. This action stimulates sensory fibers of the ilioinguinal nerve, consequently activating the motor fibers originating in the thoracolumbar segment (i.e., L1–L2) of the genital branch of the genitofemoral nerve, which causes the cremaster muscle to contract and elevate the testis. Like other superficial reflexes, it is simply graded as present or absent. The absence of cremasteric reflex suggests an upper motor neuron lesion above L1–L2.

The BCR consists of the contraction of the bulbocavernosus muscle in response to squeezing the glans penis (or clitoris), and is mediated through the pudendal nerve. In case of a complete lesion, the presence of BCR is indicative of intact S2–S4 spinal reflex arcs and loss of supraspinal inhibition, determining an upper motor neuron lesion; its absence indicates a lower motor neuron lesion. The BCR further helps distinguish conus medullaris from cauda equina syndromes [31].

The eBCR was recorded from the bulbocavernosus muscles using concentric needle electrodes and elicited by delivering brief electric shocks (square-wave of 200 µs pulse-width) through a bipolar direct-current electric stimulator (Nihon Kohden; Tokyo, Japan), equipped with ring-electrodes (placed one at the corona and one approximately 3 cm proximal to the corona). In this way, it is possible to explore the neural pathway involving the dorsal nerve of the penis-pudendal nerve and Onuf’s nucleus (S2–S4), through which pelvic muscles can dramatically increase penile rigidity of an erect penis. The stimulation intensity was seven times the individual’s sensory threshold, which is sufficient to ensure a steady reflex response [31,32]. The scanning time was 5 ms/division, bandwidth range was 10 Hz to 2 kHz, and persistence time was 100 ms. Electrode impedance was kept always <5 kΩ. We measured the latency (calculated based on the beginning of the stimulus and the start of reflex response) and peak-to-peak amplitude of eBCR. Latency reflects the integrity of the entire afferent and efferent arcs of the BCR. Amplitude reflects the increased or reduced excitability of the medullaris neurons in the epiconus. This may be useful to demonstrate an association of upper and lower motor neuron lesions in the lower sacral segments, which may be sometime difficult to demonstrate clinically. eBCR response was rated as abnormal if the latency exceeded 3 times the standard deviations from the mean (about 40 ms).

The integrity of the sensory pathway from penis to brain cortex (including dorsal nerve of penis, pudendal nerve, sacral/lumbar spinal cord) was assessed by measuring PSEP latency and amplitude. The former reflects the integrity of the entire afferent pathway from penis to cortex. The latter is a general marker of increased or reduced cortical excitability. The PSEP responses were recorded by using the same stimulation setup as per eBCR test (Figure 3) [33]. The recording was done with surface electrodes placed in the midline of the scalp, 2 cm behind the vertex region. A reference electrode was placed in the midline of the forehead at the Fz region according to the 10–20 International System (Figure 3). The intensity of the stimulus was three times the individual’s sensory threshold. The frequency of the square wave was 5 pulses/s and was averaged across 200 waves, obtaining a P41 wave. The scanning time was set to 0.2 ms/division, and the relevant persistence time was 100 ms, with a bandwidth ranging from 10 Hz to 5 kHz. Electrode impedance was kept always <5 kΩ.

Emotional status was evaluated by the Hamilton Depression Rating Scale (HDRS), a 17–21-item scale measuring the severity of depressive and somatization symptoms, where a score of ≥15 is generally regarded as indicative of a diagnosis of depression.

All patients were assessed at baseline (T0), after the treatment (T1), and three months after the end of the MV protocol (T4). 

### 2.4. Statistical Analysis

First, the Kolmogorov–Smirnov test and Shapiro–Wilk test were used to assess the normality of distribution and the homogeneity of variance of the data, respectively (all *p* > 0.2). A one-way ANOVA was used to analyze the differences in clinical and electrophysiological parameters over time. A value of *p* ≤ 0.05 was considered statistically significant. A Bonferroni correction for multiple comparison was applied. The comparison of eBCR and PSEP parameters was performed using a Kruskal–Wallis rank. The datasets used and/or analyzed during the current study are available from the corresponding author on reasonable request.

### 2.5. Ethical Approval

All procedures performed in studies involving human participants were in accordance with the ethical standards of the institutional and/or national research committee and with the 1964 Helsinki declaration and its later amendments or comparable ethical standards. The Local Ethics Committee approved the present study (IRCCSME ID: 32/2017). Informed consent was obtained from all individual participants included in the study.

## 3. Results

All patients enrolled in the study were affected by incomplete SCI (ASIA C-D). The patients had some sensory and motor preservation (C) or useful motor function (D). Specifically, a patient would be an ASIA C if more than half of the muscles evaluated had a grade of less than 3/5 on the Muscle Research Council scale. If not, the person would be an ASIA D. All patients complained of mild/moderate to severe ED, as per IIEF score, with regard to erectile and orgasmic functions. None of the patients complained of significant depressive symptoms (HDRS < 15). Further, four of them complained of urinary incontinence. All patients presented with weak or absent BCR, absent monolateral or bilateral cremasteric reflex, lack or presence of sensation, mild-to-moderate spasticity (MAS 1–2) and mild pain (VAS 1–3) of pelvic floor muscles, marked reduction of eBCR and PSEP amplitude and a clear increase of PSEP latency. eBCR latency was within the normal range. The clinical-demographic characteristics of the samples are reported in Table 1.

All patients well tolerated MV, and completed the protocol without any adverse event. All participants showed a significant improvement in IIEF total score (F_(2,18)_ = 88, *p* <0.0001), erectile (F_(2,18)_ = 65, *p* < 0.0001) and orgasmic function (F_(2,18)_ = 78, *p* < 0.0001) (Figure 4). All significant changes were present at both T1 and T4 compared to baseline (T0) (Figure 4).

MAS and subjective pain assessment (VAS) significantly decreased (F_(2,18)_ = 49 *p* < 0.0001, and F_(2,18)_ = 12 *p* < 0.0001, respectively). All patients with weak BCR and cremasteric reflex showed a better clinical response. The patients were labeled as “with preserved sensation” if all the stimuli provided to the six groin and penile sectors with the monofilament were experienced all three times correctly. Even though six-out-of-ten patients did not get “preserved sensation”, they experienced sensory stimuli all three times correctly in 3/4-out-of-six sectors, on average.

Concerning the electrophysiological measures, we found a significant increase in eBCR (F_(2,18)_ = 7.8, *p* = 0.004) and PSEP amplitude (F_(2,18)_ = 9, *p* = 0.002). The latency of eBCR and PSEP did not vary. All these changes were significant at both T1 and T4 compared to the baseline (T0) (Figure 4).

## 4. Discussion

To the best of our knowledge, this is the first attempt to improve ED by means of focal MV to pelvic muscles in men with SCI, since previous works applied whole body vibration to potentiate weak pelvic floor muscles and reduce urinary incontinence [19,20,21,22,23,24,25,26,27].

The most significant finding of our feasibility study consisted of the fact that the patients reported an improvement in erection during sexual activity, inter-course completion, and reaching ejaculation and orgasm (as per IIEF). A reduction in pelvic floor muscle spasticity and pain (as per MAS and VAS), and better reflexive responses were also found, besides a significant amplitude increase in eBCR and PSEP. Clinical changes lasted up to three months.

It is likely that the improvement in ED depended on MV-induced reduction of spasticity. However, spasticity of the muscles targeted by MV (including the bulbocavernosus, ischiocavernosus, external sphincter, and superficial transverse perineal) can regularly enhance the rigidity of erections and the veno-occlusive mechanism [34,35,36].

It is hypothesizable that MV may potentiate the mechanisms of reciprocal innervation through the presynaptic inhibition at segmental level, consequently improving muscle synergies and counterbalancing the segmental hyperreflexia [37,38,39,40,41], as suggested by the increase in eBCR amplitude. MV after effects may also depend on supraspinal mechanisms. In fact, focal MV can modify sensorimotor cortex excitability [42,43,44], as suggested by the PSEP amplitude increase. This may in turn contribute to reduce segmental hyper-excitability by harnessing the abovementioned mechanisms of reciprocal innervation. Studies employing transcranial magnetic stimulation demonstrated that focal MV increases or decreases motor evoked potential amplitude and short intracortical inhibition strength in the vibrated muscles, while opposite changes occur in the neighboring muscles [33,45,46,47]. In this way, pelvic MV may contribute to regulate the contraction and excitability dynamics of the pelvic floor muscles involved in erection. Sensory inputs from penile skin, prepuce and glans conveyed by the dorsal penis nerve may also contribute to improve erection [22,25], given that we vibrated the lower trunk muscles, which partially contribute to sexual function dynamics. In addition, the improvement in erection may depend on the effects of MV on the specific properties of the muscles and surrounding connectivity tissues (including viscoelasticity), as well as on vessel vasodilatation. Vibrations can stimulate the release of neurotransmitters and nitric oxide (NO) from the parasympathetic and non-adrenergic non-cholinergic fibers of cavernous nerve terminals of the pudendal nerve, thus evoking a reflexogenic erection [3,8,9,15,25].

To summarize, MV may contribute to reduce segmental spasticity and improve muscle synergies, thus improving erection, through both bottom-up (i.e., sensory inputs resetting sensorimotor hyperexcitability) and top-down mechanisms (re-afferent descending volleys from sensorimotor cortex to spinal centers).

As additional findings, patients reported some improvement in genital sensation, even though this did not reach statistical significance. Further assessment is mandatory to verify whether MV may represent a non-surgical modality to improve sensation in neurological patients [23]. Moreover, four patients reported urinary continence improvement. As we did not specifically address this issue, we can only speculate that the contemporary use of pen- and cup-like probes to stimulate the perineum and the lower trunk muscles allowed the deep floor muscles to be reached, which provide support for pelvic organs, urinary continence, and intestinal emptying.

## 5. Limitations and Conclusions

There are some limitations in the study to acknowledge. First, the small sample size does not let us generalize our results. However, this is a feasibility study intended to prove the safety and the potential efficacy of MV in improving ED following SCI. Further studies with a larger sample size, including females, should be conducted to confirm the benefits of MV. Second, since improvement in erection may be ascribed to psychogenic arousal and not solely to reflexogenic mechanisms, the effectiveness of MV needs to be confirmed by comparing real and sham MV. Third, as the patients were followed only up to three months after the end of the treatment, studies with a longer follow-up period are needed to establish whether and to what extent the MV after effects last. Last, since patients’ inclusion criteria may have influenced MV outcomes, we enrolled only men with incomplete SCI. It would be interesting to use this protocol also in individuals with complete SCI and in those with lesions involving the sacral spinal segments.

Notably, we excluded from this study patients taking erectogenic aids, nutraceutics, antispastics, and psychoactive drugs. This was necessary to maintain a homogeneous group and avoid confounding effects due to these drugs. Antispastic and erectogenic aids naturally interfere with the neurophysiological and vasculogenic mechanisms triggered by MV. For instance, baclofen can itself cause ED, probably by over-stimulating the central and spinal inhibitory mechanisms overseeing erectile function [9,48]; PDE-5 inhibitors may interfere with MV due to the common effect on NO release from nerve terminals and endothelial cells in the corpus cavernosum. Many nutraceutics target bodily vasculogenic and metabolic mechanisms, potentially altering the erectile processes when coupled with MV. Side effects of some psychoactive drugs may impact erection, biasing MV effects [49]. Thus, other studies are necessary to better clarify the neurophysiological basis of MV effects concerning ED treatment, also by testing MV-drug interaction and NO blood levels.

In conclusion, pelvic floor MV seems promising to improve ED in men with incomplete SCI. Given that we designed a pilot study, larger sample randomized clinical trials, including additional neurophysiological measures, are necessary to confirm MV as an add-on treatment to the conventional ED pharmacological, rehabilitative, and counseling approaches.

## Figures and Tables

**Figure 1 jcm-08-00658-f001:**
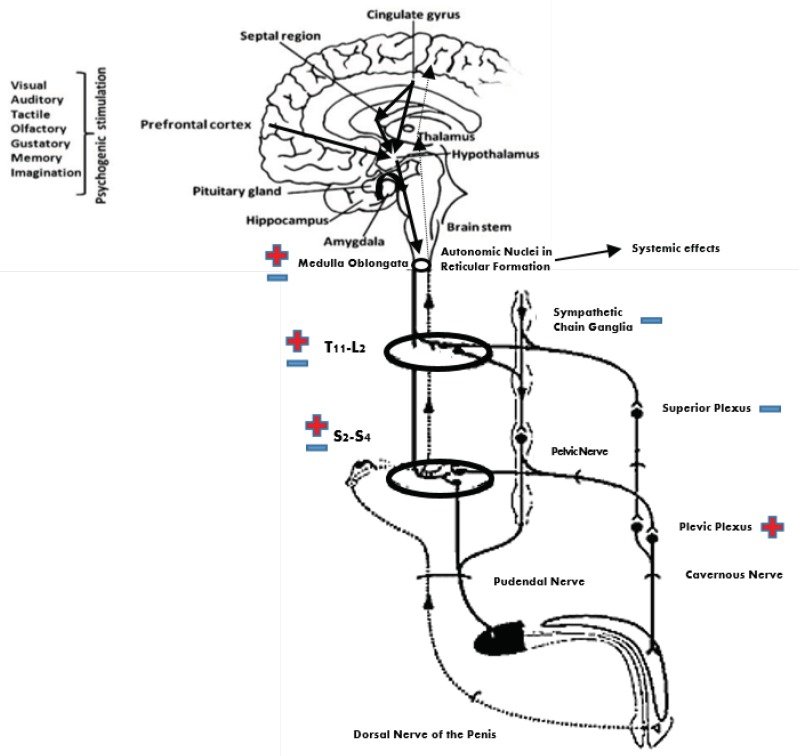
Diagram of the nervous pathways regulating erection. Reflexogenic erection depends on a circuitry made of the dorsal nerve of the penis/pudendal nerve, which conveys sensory input, a parasympathetic center in the sacral spinal cord, and the pelvic plexus/cavernous nerve, which convoys efferent output to the penis. Sensory input reaches the brain via ascending spinal pathways. The brain can initiate psychogenic erection by various stimuli received by or generated within the brain. It also controls lumbar sympathetic and sacral parasympathetic outflow to the penis. Overall, there is one penile tumescence pathway (lumbar and sacral spinal segment → pelvic plexus → cavernous nerve → penis) and three penile detumescence pathways (1: lumbar spinal segment → superior plexus → cavernous nerve → penis; 2: lumbar spinal segment → paravertebral sympathetic chain → pelvic/cavernous nerve → penis; 3: lumbar spinal segment → paravertebral sympathetic chain → pudendal nerve → penis). + and − indicate penile tumescence or detumescence effect, respectively.

**Figure 2 jcm-08-00658-f002:**
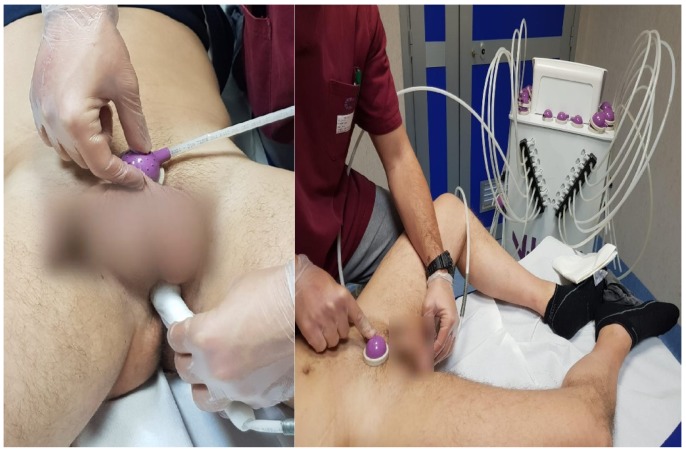
Placement of the suprapubic and perineum probes prior to body fixation by Velcro-straps.

**Figure 3 jcm-08-00658-f003:**
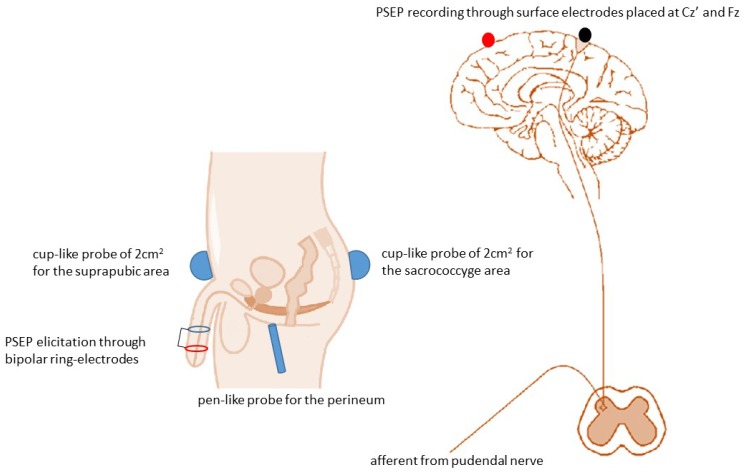
Placement of the probes and pudendal nerve somatosensory-evoked potential (PSEP) setup.

**Figure 4 jcm-08-00658-f004:**
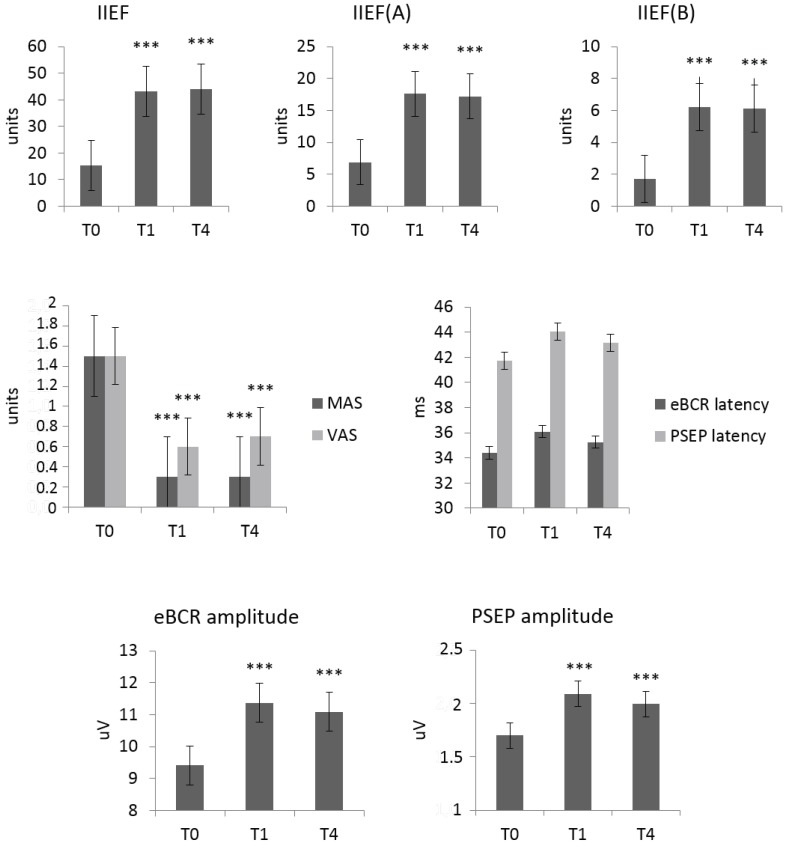
Clinical and electrophysiological aftereffects of pelvic muscle vibration at T0 (baseline), T1 (post-treatment) and T4 (3 months after the end of the treatment). Significance (*) was tested between T1 and T0 and T4 and T0 (*** *p* < 0.001). Vertical bars indicate s.d. IIEF—International Index of Erectile Function (A—erectile function, B—orgasmic function); eBCR—electrophysiological bulbocavernosus reflex; MAS—Modified Ashworth Scale; VAS—visual Analogue Scale for pain esteem; PSEP—pudendal nerve somatosensory-evoked potential.

**Table 1 jcm-08-00658-t001:** Clinical-demographic characteristic of the sample at baseline.

Age (y)	dd (m)	SCI Level	ASIA Level	BCR	eBCR		Cremasteric Reflex	Sensory Preservation	PSEP		MAS	VAS	IIEF		
					amp (μV)	lat (ms)			lat (ms)	amp (μV)			Total	A	B
65	8	T12–L1	D	no	5	34	present/asymmetric	yes	48	3	2	2	16	2	1
38	10	L1–L2	D	weak	6	35	present/asymmetric	no	/	0	1	0	18	6	2
62	12	T9–D12	C	no	6	34	absent	no	44	1	1	2	17 *	5	2
26	14	T10–L1	C	yes	11	34	present/asymmetric	yes	41	3	2	2	18	8	3
45	6	L1–L2	D	no	6	36	present/asymmetric	yes	43	4	2	3	13 *	9	0
48	10	L1–L3	C	no	8	33	absent	no	39	2	1	0	16 *	9	1
45	4	L1–L3	C	weak	11	36	present/asymmetric	no	38	1	1	2	14	12	2
22	13	T12–L2	C	no	4	34	absent	no	/	0	1	1	14	5	3
39	3	T12–L1	D	no	5	34	present/asymmetric	yes	36	3	3	3	11	6	3
48	12	T8–D10	C	weak	8	34	absent	no	42	1	1	0	18 *	7	0

Legend: SCI—spinal cord injury; ASIA—American Spinal Injury Association; * urinary incontinence; BCR—bulbocavernosus reflex; eBCR—electrophysiological bulbocavernosus reflex; MAS—Modified Ashworth Scale; VAS—visual Analogue Scale; IIEF—International Index of Erectile Function; PSEP—pudendal nerve somatosensory-evoked potential.

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
