# Peer review of "Improving Sexual Function by Using Focal Vibrations in Men with Spinal Cord Injury: Encouraging Findings from a Feasibility Study"

_jcm, 2019, doi:10.3390/jcm8050658_

Round 1
Reviewer 1 Report
Very interesting study promising hope for SCI patients. I look forward to bigger patient data with controls.
line 118 do you mean using instead of busing?
284 pilot study
Author Response
We thank the reviewer for the appreciation for our manuscript and for the useful comments to improve its quality.
1) Very interesting study promising hope for SCI patients. I look forward to bigger patient data with controls.
The Aim of this pilot study was to evaluate the feasibility and efficacy of pelvic MV in improving ED in men with SCI. We agree on the fact that a larger number of patients is needed to confirm our data. This issue has been added in the limitation and future perspective section.
2) line 118 do you mean using instead of busing?
Corrected in using.
3) 284 pilot study
Corrected.
Reviewer 2 Report
The present study investigated whether the focal muscle vibration may be applied to pelvic muscle to treat erectile dysfunction in adult men in the chronic state of spinal cord injury. Even though I found the study extremely interesting and very much needed, the way it is written shut my optimist view of it. The results and the methods section are well presented and described. However, the rest of the paper should undergo a major rewriting process.
I wrote here the major concerns and you will find more information about it and minor suggestions in the attached file.
Major concerns:
-The abstract, introduction and discussion sections are extremely difficult to read with several points not well explained. Often sentences do not have a proper structure and the reader has to make continuous efforts to interpret what the authors want to say. More in detail:
-in the abstract, unnecessary information should be avoided. It implies the deletion of sentences in brackets and p values. The sentences in line 19 and 23 should be rephrased since it is not well expressed.
-Introduction: Even though I understood the context the authors want to give to the reader, the introduction does not show a clear path to follow and it mentions several topics that are not logically and directly linked to each other. The authors should make it more straightforward. First, they should decide the logic (e.g. ED→features→relationship with spasticity→current therapies→MV. Next, they should avoid all unnecessary and redundant information.
-discussion: I personally suggest a massive rewriting of the discussion, making each paragraph simple and easy to read. The use of more reference is strongly suggested since most of the interpretations of this section are not supported by that.

Author Response
We thank the reviewer for the appreciation for our manuscript and for the useful comments to improve its quality. The manuscript was rewritten in keeping with reviewer’s suggestions and the comments reported in the attached file.
1) The abstract, introduction and discussion sections are extremely difficult to read with several points not well explained. Often sentences do not have a proper structure and the reader has to make continuous efforts to interpret what the authors want to say.
The manuscript was entirely revised to make the sentences more clear, also adding the relevant information where missing.
2) More in detail:
a. in the abstract:
i. unnecessary information should be avoided. It implies the deletion of sentences in brackets and p values.
Done.
ii. The sentences in line 19 and 23 should be rephrased since it is not well expressed.
Checked and corrected
b. Introduction: Even though I understood the context the authors want to give to the reader, the introduction does not show a clear path to follow and it mentions several topics that are not logically and directly linked to each other. The authors should make it more straightforward. First, they should decide the logic (e.g. ED→features →relationship with spasticity→current therapies→MV. Next, they should avoid all unnecessary and redundant information.
The logical path of introduction was completely revised according to the useful reviewer’s suggestion.
c. discussion: I personally suggest a massive rewriting of the discussion, making each paragraph simple and easy to read. The use of more reference is strongly suggested since most of the interpretations of this section are not supported by that.
The discussion was entirely revised to make the sentences more clear, also adding the relevant information and references where missing.

Reviewer 3 Report
Calabrò et al. study entitled "Improving sexual function by using focal vibrations in men with spinal cord injury: Encouraging findings from a feasibility study" is an important question and necessity to the spinal cord injury patients. Authors try to design a study and use a non-invasive tool to improve the ED in SCI patients. There are several concerns about the study and they are mentioned below as
Authors did not mention about any results of their electrophysiology study in the abstract and presented the data in figure 1 and in result section which are not matching.
Figure 1, there is a need to label the y-axis in all the graph as well as properly arrange the significant signs. Figure legend needs a more detailed description of all these graphs.
The manuscript needs to simplify throughout by taking care of abbreviations and too much text in parenthesis.
There is a clear cut lack of control groups in the study using either PDE-5 as a positive control as well as sham control for MV.
Authors mentioned PDE-5 first time in the discussion but did not include in the study. Why?
The schematic or real figure for MV stimulation should be added in the manuscript along with the PSEP recording graph. MV application full details as schematic should be mentioned in the paper.
Authors hypothesize throughout the study as you see about the MV effect on NO release a related mechanism. Why the authors did not show in their study about the NO mechanism related to MV effects on ED? Authors should do this in the blood.
Minor comments
Abstract-Spinal cord injury should be in lower case.
Abstract-Put a period after follow-up period in line 23 and correct the sentence after that.
Re-write the 1st paragraph of the introduction.
What is SD?
Correct table 1 for "yesasym".
Explain the T4 in the beginning.
Author Response
We thank the reviewer for the useful comments to improve the quality of our manuscript. The manuscript was rewritten in keeping with reviewer’s suggestions and the comments reported in the attached file.
1- Authors did not mention about any results of their electrophysiology study in the abstract.
We added the missing information
2- the data in figure 1 and in result section are not matching.
Checked and corrected for incongruences.
3- Figure 1, there is a need to label the y-axis in all the graph as well as properly arrange the significant signs. Figure legend needs a more detailed description of all these graphs.
Corrected as per reviewer’s suggestion.
4- The manuscript needs to simplify throughout by taking care of abbreviations and too much text in parenthesis.
Checked and corrected.
5- There is a clear cut lack of control groups in the study using either PDE-5 as a positive control as well as sham control for MV.
As correctly argued by the reviewer, the improvement in erectile response may be not solely based on reflexogenic mechanisms. We have now more clearly stated that the actual effectiveness of MV in the improvement of ED has to be confirmed by comparing real and sham MV. Further, we excluded patients taking erectogenic aids to maintain a uniform patient group and avoid confounding vasculogenic effects due to the intake of the drugs potentially interfering with MV effects. Thus, other studies are necessary to better clarify the neurophysiological basis of MV effects concerning ED treatment.
6- Authors mentioned PDE-5 first time in the discussion but did not include in the study. Why?
Accordingly, PDE-5 were first introduced in the introduction section. Then, we toned down the argumentation on PDE-5in the discussion since redundant. We excluded patients taking erectogenic aids to maintain a uniform patient group and avoid confounding vasculogenic effects. Thus, other studies are necessary to better clarify the neurophysiological basis of MV effects concerning ED treatment (see also point 5).
7- The schematic or real figure for MV stimulation should be added in the manuscript along with the PSEP recording graph. MV application full details as schematic should be mentioned in the paper.
We added the missing information.
8- Authors hypothesize throughout the study as you see about the MV effect on NO release a related mechanism. Why the authors did not show in their study about the NO mechanism related to MV effects on ED? Authors should do this in the blood.
We thank the reviewer for this interesting suggestion. Unfortunately, we did not test NO in the blood. In keeping with these issues, we toned down the argumentation on PDE-5 and NO in the discussion, since redundant. The lack of NO testing in the blood was added as study limitation.
9- Minor comments
a. Abstract-Spinal cord injury should be in lower case.
Corrected.
b. Abstract-Put a period after follow-up period in line 23 and correct the sentence after that.
The paragraph was entirely revised to make the sentences more clear.
c. Re-write the 1st paragraph of the introduction.
The paragraph was entirely revised to make the sentences more clear.
d. What is SD?
Corrected, it was ED
e. Correct table 1 for "yesasym".
Corrected.
f. Explain the T4 in the beginning.
Done.
Round 2
Reviewer 3 Report
Please check for grammar and other correction before final publication.
Figure 3 needs to improve by fixing the asterisk signs.
Thanks
Author Response
We thank the reviewer for the appreciation to our manuscript and the useful comments to improve its quality.
1- Please check for grammar and other correction before final publication.
The manuscript was extensively revised for grammar and other corrections.
2- Figure 3 needs to improve by fixing the asterisk signs.
Checked and corrected.
Kindest regards,
The authors